# Hepatic Artery Delineation on Ultrasound Volumes Comparing B-Flow and Color Doppler for Postoperative Monitoring of Pediatric Liver Transplants

**DOI:** 10.3390/diagnostics14060617

**Published:** 2024-03-14

**Authors:** Elena Dammann, Leonhard Steinmeister, Michael Groth, Uta Herden, Lutz Fischer, Florian Brinkert, Jan Beime, Magdalini Tozakidou, Peter Bannas, Jochen Herrmann

**Affiliations:** 1Section of Pediatric Radiology, Department of Diagnostic and Interventional Radiology and Nuclear Medicine, University Medical Center Hamburg-Eppendorf, 20251 Hamburg, Germany; 2Department of Radiology, Alta Klinik Bielefeld, 33602 Bielefeld, Germany; 3Radiologie Vechta, Marienstraße 6-8, 49377 Vechta, Germany; 4Department of Visceral Transplantation, University Medical Center Hamburg-Eppendorf, 20251 Hamburg, Germany; 5Department of Pediatrics, University Medical Center Hamburg-Eppendorf, 20251 Hamburg, Germany; 6Department of Diagnostic and Interventional Radiology and Nuclear Medicine, University Medical Center Hamburg-Eppendorf, 20251 Hamburg, Germany

**Keywords:** B-flow sonography, Color Doppler sonography, hepatic artery, pediatric liver transplantation, vascular complications

## Abstract

(1) Background: Accurate hepatic artery (HA) depiction following pediatric liver transplantation (LT) is essential for graft surveillance but challenging on ultrasound (US). This study assesses if improved HA delineation can be achieved by recording two-dimensional US volumes in Color Doppler (CD) and B-flow technique. (2) Methods: Of 42 consecutive LT, 37 cases were included, and HA delineation was retrospectively rated using a four-point score (0 = HA not detectable, 3 = HA fully detectable, separable from portal vein) within 48 h post-LT (U1) and before discharge (U2). (3) Results: Adding B-flow compared with CD alone showed superior results at neohilar (U1: 2.2 ± 1.0 vs. 1.1 ± 0.8, *p* < 0.0001; U2: 2.5 ± 0.8 vs. 1.5 ± 0.9, *p* < 0.0001) and segmental levels (U1: 2.8 ± 0.6 vs. 0.6 ± 0.8, *p* < 0.0001; U2: 2.8 ± 0.6 vs. 0.7 ± 0.5, *p* < 0.0001). (4) Conclusions: Standardized US volume recordings combining B-flow and CD can effectively delineate the HA along its vascular course in pediatric LT. The technique should be further evaluated as a standard monitoring instrument to rule out vascular complications after LT.

## 1. Introduction

Liver transplantation (LT) is the state-of-the-art curative treatment option for children with end-stage liver disease, showing excellent short- and long-term survival rates [1]. In children, vascular complications after LT occur at a higher rate [2,3] than in adults [4], and acute hepatic artery (HA) thrombosis is a main reason for early graft loss if not swiftly revised [5,6,7]. Therefore, accurate assessment of the graft’s vascular supply is vitally important during the postoperative phase.

Ultrasound (US) is the accepted primary imaging method to monitor children after LT [8]. However, post-transplantation US is known to be technically demanding, especially in smaller children. This is explained by complicated vascular anatomy, small-sized vessels with caliber mismatch at the anastomosis, high heart rates, and a small acoustic window due to secondary abdominal wall closure in an intensive care setting [2,3]. Color Doppler (CD) is widely used for post-transplant evaluation, but the method has limitations in depicting smaller and low-flow vessels. CD regularly overestimates true vessel size due to blooming artifacts, potentially obscuring detection of the HA in proximity to the portal vein by spatial overlap [9,10,11]. As an alternative, different non-Doppler-based flow imaging methods have been developed and included by vendors on their US platforms. Generally, these applications are characterized by higher spatial resolution, higher frame rate capabilities, and abilities to detect slow-flow vessels [12,13]. Among these is B-flow, which is based on the subtraction of received amplitudes of grayscale US, enabling the direct visualization of flowing blood cells by boosting weak motion signals and suppressing stationary tissue signals. It is less angle-dependent than Doppler-based flow techniques and has been shown to be especially useful in areas with simultaneous low and high blood flow. Its efficacy in detecting small vessels has been demonstrated across diverse clinical applications, including the hepatic vasculature in native and transplanted livers [14,15,16,17]. 

As vascular complications can develop gradually and anywhere along the extra- and intrahepatic vascular course, a thorough assessment and exact documentation is necessary to achieve comparability [8]. A recent survey among European transplantation centers identified various shortcomings in the present application of US, including a lack of standardization and insufficient documentation practices [18]. Instead of digital storage of exemplary single images, the recording of standardized US volumes containing continuous, thin through-plane slices in different techniques can be performed [19]. Volume US documentation facilitates offline reading as well as image interpretation in situations with complex vascular anatomy but, to the best of our knowledge, has not yet been systematically evaluated in LT monitoring [14,20,21,22]. 

The aim of our study was to describe a method to record two-dimensional US volumes for extended standardized documentation in post-transplant US monitoring of children. We hypothesized that the US volumes can be used for secondary evaluation and that B-flow, in combination with CD, will improve the diagnostic quality of hepatic artery depiction in pediatric liver transplants. 

## 2. Materials and Methods

### 2.1. Study Population

We included pediatric patients after LT who received protocol US examinations at our institution between December 2015 and December 2016 in our retrospective study. Out of 42 consecutive transplantations performed during this period, five cases were excluded because of incomplete clinical documentation. Two patients needed re-transplantation during the study period, and both LTs were included. A total of 37 cases with LTs performed in 35 patients were included in this study (mean age at LT 6.9 ± 6.9 years; range 1 week—22 years; 17 male, 18 female). Primary diagnoses leading to transplantation were: Maple syrup urine disease (*n* = 8), biliary atresia (*n* = 7), acute hepatic failure (*n* = 6), autoimmune liver disease (*n* = 3), citrullinemia type 1 (*n* = 2), glycogen storage disease type 1 (*n* = 1), alpha1-antitrypsin deficiency (*n* = 1), hyperoxaluria (*n* = 1), progressive familial intrahepatic cholestasis type 1 (*n* = 1), argininosuccinate lyase deficiency (*n* = 1), carbamoyl phosphate synthetase1 deficiency (*n* = 1), hepatoblastoma (*n* = 1), primary sclerosing cholangitis (*n* = 1), secondary sclerosing cholangitis (*n* = 1), and liver disease of unknown cause (*n* = 2). Details regarding patient and transplant data are summarized in Table 1. 

### 2.2. Ultrasound Examination

All standardized US examinations were performed by a single pediatric radiologist (J.H.) with more than 12 years of experience in pediatric liver transplantation using a GE Logiq 9 US system (GE Medical Systems, Milwaukee, WI, USA) with a 2–5 MHz convex probe. All children received a protocol-based US examination, which included the acquisition of freehand two-dimensional volumes in the axial plane in B-mode, CD, and B-flow with identical starting and ending positions and probe movement (Figure 1). In the case of full grafts and right split grafts, the probe was positioned in the intercostal space at the right midclavicular line. In patients with left split grafts (segment 2,3), the probe was localized above the graft approximately in the midline within the upper abdomen. The volume recording started above the level of venous hepatic outflow and ended below the level of the neohilum, including the intra- and extrahepatic HA and PV. Spectral analyses of both vessels were measured at the neohilum, and maximum velocities (cm/s) as well as resistance indices (defined as: (peak systolic velocity—end-diastolic velocity)/peak systolic velocity) of the HA were extracted. Correction of the Doppler angle was performed.

### 2.3. Image Analysis

For each liver transplant, US volumes recorded at two time points were assessed (U1: within 48 h after LT; U2: before discharge, 49 ± 64 days after LT). The assessment was performed offline on an integrated Radiology Information System–Picture Archiving and Communication System (RIS-PACS) workstation (Centricity RIS-i 5, Centricity Universal Viewer, GE Healthcare, Chicago, IL, USA) in consensus reading by two radiologists with 12 years (BLINDED) and four years (BLINDED) of experience in LT US. The readers were blinded to patient information and time points of the examinations. The offline ratings were performed in random order. For each case and date, the CD volume was rated first. Afterwards, the corresponding B-flow volume was made available and rated side-by-side with the CD images. For each dataset, CD alone and B-flow with CD, the degree of HA detectability was assessed at the extrahepatic, neohilar, and segmental vessel locations according to the following four-point scoring system (Figure 2):-S 0 = HA not detectable, only PV detectable-S 1 = HA discontinuously detectable but not separable by a gap from the PV-S 2 = HA continuously detectable but not separable by a gap from the PV*or* HA discontinuously detectable but separable by a gap from the PV-S 3 = HA continuously detectable and completely separable by a gap from the PV

A total score of HA detectability was calculated by the summation of three regional scores. A full score of 9 indicated perfect vessel delineation along the entire vascular course. A total score > 3 was set as a threshold for acceptable detectability. 

Maximum PV and HA velocities (cm/s) as well as resistance indices of the HA were extracted for each time point from the original report. Vascular complications regarding the grafts’ vascular supply were noted if present.

### 2.4. Statistical Analysis

For each time point (U1 and U2), data were descriptively analyzed, including detectability scores of the HA (S 0–3) on US volumes, maximum velocities of the HA and PV, as well as resistance indices (RI). Regional detectability scores of the HA with CD and CD with B-flow were compared for time point U1 and for time point U2 using a two-tailed Wilcoxon test. Velocities and RI were compared between both time points by applying two-tailed paired sample t-tests. Statistical analysis was performed with commercially available software tools (MedCalc for Windows, version 19.4, Mariakerke, Belgium and Excel for Mac, version 16.78, Microsoft Corporation, Redmond, WA, USA). A *p*-value of less than 0.05 was considered to be significant. Data are presented as mean with standard deviation.

## 3. Results

A review of the 37 on-site US examinations directly after the transplantation (U1) and before discharge (U2) showed that there were no vascular LT complications. The peak flow velocities of the HA and PV were significantly higher at U1 than at U2 (HA, 88.2 ± 22.1 cm/s vs. 67.4 ± 31.8 cm/s, *p* < 0.01; PV, 59.5 ± 43.0 vs. 36.5 ± 17.4 cm/s, *p* < 0.001). The HA resistance index (RI) did not differ significantly between U1 and U2 (0.67 ± 0.13 vs. 0.63 ± 0.11; *p* 0.13). 

Off-site analyses of the US volumes showed superior delineation of the HA and better differentiation from the PV for B-flow together with CD than for CD alone at the neohilar level (U1: 2.2 ± 1.0 vs. 1.1 ± 0.8; *p* < 0.0001; U2: 2.5 ± 0.8 vs. 1.5 ± 0.9, *p* < 0.0001) and at the intrahepatic segmental level (U1: 2.8 ± 0.6 vs. 0.6 ± 0.8; *p* < 0.0001; U2: 2.8 ± 0.6 vs. 0.7 ± 0.5; *p* < 0.0001) (Figure 3 and Figure 4). At the extrahepatic level, the degree of HA detectability was relatively poor for both methods and without a significant difference (U1: 1.3 ± 1.2 vs. 1.2 ± 0.9, *p* = 0.4; U2: 1.2 ± 1.2 vs. 1.2 ± 1.0, *p* = 0.8). 

Combined analyses of B-flow with CD compared to CD alone resulted in a substantially better overall visibility of the HA along its course (total score at time point U1, 6.3 ± 2.2 vs. 2.8 ± 1.6; time point U2, 6.5 ± 2.1 vs. 3.3 ± 1.8). A total score >3, reflecting the threshold for an acceptable degree of HA depiction, was achieved in 89.2% (33/37 cases) on B-flow together with CD volumes compared to 29.7% for CD alone (11/37 cases).

## 4. Discussion

The current paper describes a new method for recording two-dimensional US volumes, applicable for postoperative surveillance of liver transplants in children. The study findings demonstrate effective delineation of the HA on these volumes. Combining CD with B-flow leads to improved HA delineation and thereby provides added certainty for a postoperative monitoring scheme. 

It has been shown that the acquisition of US volumes compared to still images can enhance reliability and that examination time can be significantly reduced while retaining and possibly improving diagnostic quality [24,25]. Recording of US volumes, which can be relatively swiftly performed, allows for limiting the examination time by shifting lengthy evaluations from the bedside to an offline read-out. This can especially be useful for critical ill patients and young children requiring minimal handling. Examples of clinical routines with incorporation of US volumes are volumetric neonatal brain evaluation, neonatal basal artery evaluation, femoral artery evaluation in infants, growth plate assessment in adolescence, as well as general abdominal US in adults [20,21,22,26,27,28]. For these diagnostic protocols, technical settings including probe orientation, starting and ending position in case of free-hand translational probe movement, as well as rules for later offline read-out have been defined. Although US volume recordings are also part of the diagnostic routine at some specialized centers for pediatric LT, no procedural recommendations have been reported so far [29].

After LT in children, most transplantation sites apply a strict monitoring regime to rule out vascular complications with up to three US examinations per day in the early postoperative phase [18]. Maintaining image quality and continuity beginning at the intraoperative phase and seamlessly extending to the follow-up phase is vitally important, as complications can develop anywhere along the complex vascular anatomy. With the availability of continuous thin-sliced image-stacks, these US volume recordings resemble the documentation practices principally known from other cross-sectional imaging methods like computed tomography (CT) or magnetic resonance imaging (MRI). The concept of extended documentation is to provide more imaging details to pediatric radiology consultants and transplant surgeons, which can be used for secondary offline evaluation to achieve the best follow-up comparability. Especially for experienced observers, it has been shown that high agreement and accurate assessment can be achieved with volume US recordings during offline read-out [21].

In the present study, US volumes recorded with two different flow-imaging techniques were assessed for their capacity to collect information on the grafts supplying vessels at three organ regions. We showed that CD US, which is considered the main method at most transplantation sites, could only partially delineate the HA at the neohilar and intrahepatic levels. The HA was usually not directly separable from the PV and was only identified in some sections of the vascular course by its higher flow. On CD, flow signals are known to extend beyond the true vessel boundaries (blooming artifact), which particularly occurs at high flow velocities and can limit the demarcation from neighboring structures like the portal vein [30]. In children, the combination of small recipient vessels and the temporarily increased PV flow-velocities after LT, running at a similar speed as the neighboring HA, can further hamper vessel differentiation on CD [31]. Visualization on Doppler US can also be impaired by periportal edema, vasospasms, low cardiac output, and high heart frequencies in children [32].

After similarly oriented B-flow image stacks were made available to the readers in our study, HA depiction was significantly improved. In most cases, the artery was clearly delineated and continuously detectable on the through–plane sequences at the neohilar and segmental locations. B-flow is known for its higher spatial and temporal resolution and is less angle-dependent than CD [33]. In addition, B-flow has a wider dynamic range. Despite significantly higher arterial and portal venous flow velocities of vessels in the early compared with the late postoperative examination, the quality of HA depiction on B-flow remained good at the intrahepatic and neohilar locations. High-flow vessels at the neohilum and low-flow vessels at the intrahepatic location can be imaged with one preset, making the technique suitable for high-resolution volume recording [12,21]. At the extrahepatic level, HA detectability was relatively poor for CD and B-flow. It is known that B-flow is prone to sound beam attenuation that increases with depth. However, when evaluating smaller body sized children, this potential limitation seems less relevant for depicting intrahepatic and neohilar vessels.

Our results of improved vessel delineation in pediatric liver transplants are in accordance with initial findings in adult patients assessing the hepatic vasculature and hypervascular hepatic tumors with B-flow, as well as detailed kidney transplant evaluation in children [16,34,35,36]. Similar findings have been reported for superb microvascular imaging (SMI), which shares advantages with B-flow. SMI has been evaluated for HA detection in pediatric LT and has shown to improve the accuracy of HA thrombosis detection in some cases [13,14]. Contrast-enhanced ultrasound (CEUS) has proven to have high accuracy in detecting vascular complications after LT. In cases of insufficient HA visualization with B-mode and CD, CEUS has shown potential as a problem-solving tool to rule out or confirm HA occlusion. Additionally, CEUS also provides additional information regarding effective parenchymal perfusion, e.g., to differentiate vascular stenosis leading to downstream perfusion deficit [31]. However, CEUS is more invasive and remains off-label for use in pediatric LT monitoring [37,38]. 

Our study has the following limitations: (1) US volumes recorded at the bedside were evaluated as a standalone offline method to assess the quality of the grafts supplying vessels. In clinical circumstances, more information would be available for secondary evaluation (including spectral waveform analysis with measurements of flow velocities and resistive indices), further enhancing US diagnostic capacity. (2) CD volumes were always rated before B-flow volumes, possibly adding bias to the scores. However, B-flow operates with full background suppression and without encoding flow direction. The reduced, angiography-like images need to be combined with CD recordings, which provide crucial information on flow direction and velocity and simultaneously offer details on liver parenchyma and surrounding anatomy. (3) The study design is retrospective, and therefore prevents assessment of the impact on clinical management. (4) All examinations recorded in pediatric liver transplants were without vascular complications, thus precluding assessment of diagnostic accuracy.

## 5. Conclusions

Our results underline the added value of combining Doppler with non-Doppler-based flow imaging techniques for postoperative monitoring protocols in children after LT. When combined with CD, the availability of B-flow significantly enhances HA detectability in children. Therefore, B-flow or another flow-imaging technique sharing similar properties, like SMI, should be considered as primary non-invasive alternatives to CD when the hepatic artery cannot be sufficiently visualized. Prospective studies should test whether combining two-dimensional CD and B-flow US volumes, as described in this study, can lead to a reduction in false-positive suspected diagnoses and thus potentially reduce unnecessary invasive examinations such as CT angiography or surgical revision.

## Figures and Tables

**Figure 1 diagnostics-14-00617-f001:**
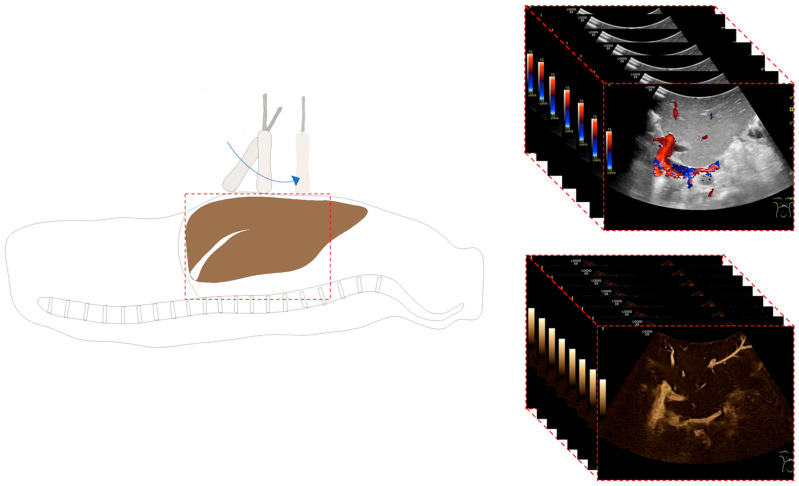
Illustration of the ultrasound (US) examination of a liver transplant. To record two-dimensional US volumes, a convex probe is placed transversally and moved slowly, following a fan-like course from the hepatic outflow towards the lower end of the graft, crossing the neohilum. Similar recordings are produced in B-mode (not shown), Color Doppler, and B-flow.

**Figure 2 diagnostics-14-00617-f002:**
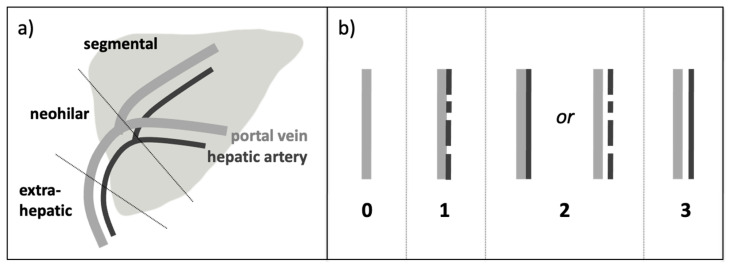
Degree of detectability of the hepatic artery (HA): (**a**) three different vessel locations: extrahepatic, neohilar, and segmental (exemplarily shown on a left-lateral split transplant). (**b**) Scoring system: (0) HA not detectable, (1) HA discontinuously detectable, not separable from the portal vein (PV), (2) HA discontinuously detectable, separable from the PV, or HA continuously detectable, but not separable from the PV; (3) HA continuously detectable and completely separable from the PV.

**Figure 3 diagnostics-14-00617-f003:**
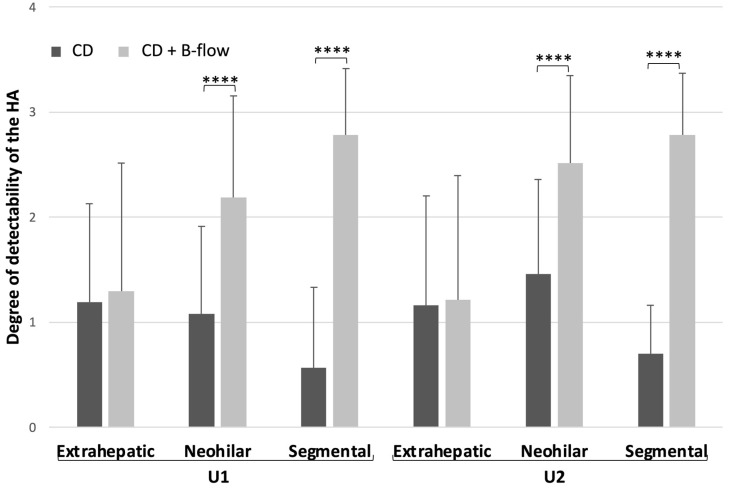
Comparison of the degree of detectability of the hepatic artery (HA) by Color Doppler (CD) and CD with B-flow at two different time points (U1: within 48 h after LTX; U2 before discharge) and for three different vessel locations (extrahepatic, neohilar, and segmental). All data are means + standard deviations. **** statistically significant.

**Figure 4 diagnostics-14-00617-f004:**
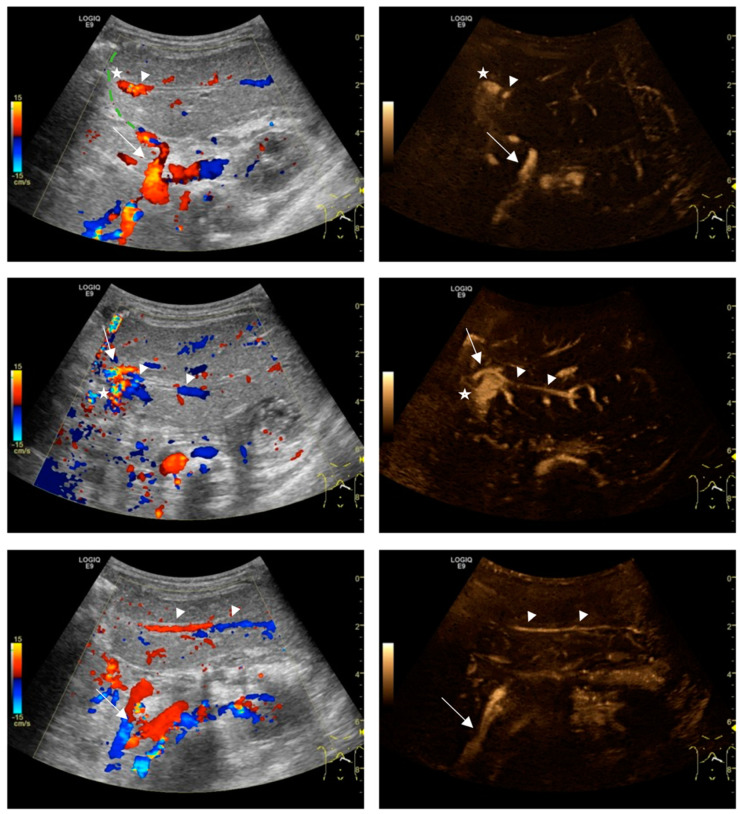
Four-month-old girl with biliary atresia and situs inversus after left-lateral split liver transplantation. Single ultrasound images were extracted from two-dimensional volumes recorded on day 1 after transplantation in Color Doppler (CD, **left** column) and B-flow (**right** column). The CD images provide information on flow direction, flow velocity, and transplant anatomy. The split surface is marked with a dashed green line (**top left**). The hepatic artery (HA) can be seen at its extrahepatic location (**top** row, arrows) and neohilar location (**middle** row, arrows). On CD, differentiation of the HA (arrow) from the portal vein (asterisks) is obscured by aliasing and blooming artifacts. On B-flow images with full background suppression, the intrahepatic HA is more clearly and continuously visible entering segment 2 (arrowheads, **upper** and **middle** row) and segment 3 (arrowheads, **bottom**). Complete volume recordings can be found in the Appendix A.

**Table 1 diagnostics-14-00617-t001:** Patient and transplant data.

Parameter	
Age at liver transplantation (years)	6.9 ± 6.9
Sex	
Number (no.) of males	17
No. of females	18
Weight at liver transplantation (kg)	25.2 ± 22.2
Preoperative patient status	
No. with high urgency	20
No. with chronic disease	17
Type of liver transplantation	
No. with orthotopic, full graft	8
No. with reduced graft	1
No. with split graft, deceased donor	21
No. with split graft, living donor	7
Liver transplantations	
No. with first	32
No. with second	5
Ischemic time (min)	
Cold	621.1 ± 176.6
Warm	36.1 ± 16.9

All data are means ± standard deviations, except where otherwise indicated. Preoperative patient status was defined according to the Eurotransplant liver allocation system, differentiating high urgency listings from chronic disease (elective) [23].

## Data Availability

Data presented in the study are available on request from the corresponding author. Restrictions apply due to privacy and legal concerns.

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
