# Peer review of "Hepatic Artery Delineation on Ultrasound Volumes Comparing B-Flow and Color Doppler for Postoperative Monitoring of Pediatric Liver Transplants"

_diagnostics, 2024, doi:10.3390/diagnostics14060617_

Round 1

Reviewer 1 Report

Comments and Suggestions for Authors

This manuscript was titled as 'Hepatic artery delineation on ultrasound volumes comparing 2 B-flow and color Doppler during postoperative monitoring of 3 pediatric liver transplants'. 

It presented a new method for recording two-dimensional US volumes, applicable for postoperative surveillance of liver transplants in children.

This good findings can provide a new concept for further studies of US usage in liver transplantation.

Author Response

Thank you!

Reviewer 2 Report

Comments and Suggestions for Authors

The present study demonstrates a new valuable ultrasound approach for surveillance of liver transplants in children.  It is well perceived and presented. I have only a few comments for the authors consideration:

L. 79: In contrast to later data, it is stated that only 35 of 37 cases had a liver transplantation? Correct?

Table 1 and elsewhere: Too many decimals.

 L. 123: "49-64" (?).

L. 129-130. The blinding is quite essential. Therefore, is it possible to better describe whether the ratings of CD alone vs CD+B flow were made randomly / blinded in a given patient / or made simultaneously in this patient.

L. 159: explain "resistance indices".

L. 164: Were p-values one- or two-tailed? Also, write that data are presented with SDs.

Fig. 3: "B-flow" should be "B-flow with CD".

L. 280: Has the abbreviation SMI been explained? 

Author Response

Thank you very much for your revision. Please see the attachment.

Reviewer 3 Report

Comments and Suggestions for Authors

The paper describes a protocol on hepatic artery delineation on ultrasound volumes.

It emphasizes the use of B-flow (BF) and color Doppler (CD) during postoperative monitoring of pediatric liver transplants, comparing the results of CD with the combined result of BF and CD.

Two measurement times and three locations are taken in consideration, and the results, covered by statistical assessment, show the superiority of CD+BF over CD alone.

The paper is well written and clearly presented. The results are covered by the methodology.

Minor editing should be done, like removing the numbering in the abstract, and adding a “to” between 49 and 64 on line 123, etc.

Having all the volumes publically available (not only two) would bring a lot to the scientific community, but it is at the author's choice.

Author Response

(The authors gave the same response as above.)
